# Macrophage Polarization Status Impacts Nanoceria Cellular Distribution but Not Its Biotransformation or Ferritin Effects

**DOI:** 10.3390/nano13162298

**Published:** 2023-08-10

**Authors:** Uschi M. Graham, Alan K. Dozier, David J. Feola, Michael T. Tseng, Robert A. Yokel

**Affiliations:** 1Pharmaceutical Sciences Department, College of Pharmacy, University of Kentucky, Lexington, KY 40536-0596, USA; graham@topasol.com; 2National Institute of Occupational Safety and Health (NIOSH), Cincinnati, OH 45213-2515, USA; xlh5@cdc.gov; 3Pharmacy Practice and Science Department, College of Pharmacy, University of Kentucky, Lexington, KY 40536-0596, USA; david.feola@uky.edu; 4Anatomical Sciences and Neurobiology, University of Louisville, Louisville, KY 40202, USA

**Keywords:** analytical microscopy, bioprocessing, cerium phosphate, iron biomineralization, macrophage phenotype, nanotoxicology

## Abstract

The innate immune system is the first line of defense against external threats through the initiation and regulation of inflammation. Macrophage differentiation into functional phenotypes influences the fate of nanomaterials taken up by these immune cells. High-resolution electron microscopy was used to investigate the uptake, distribution, and biotransformation of nanoceria in human and murine M1 and M2 macrophages in unprecedented detail. We found that M1 and M2 macrophages internalize nanoceria differently. M1-type macrophages predominantly sequester nanoceria near the plasma membrane, whereas nanoceria are more uniformly distributed throughout M2 macrophage cytoplasm. In contrast, both macrophage phenotypes show identical nanoceria biotransformation to cerium phosphate nanoneedles and simultaneous nanoceria with ferritin co-precipitation within the cells. Ferritin biomineralization is a direct response to nanoparticle uptake inside both macrophage phenotypes. We also found that the same ferritin biomineralization mechanism occurs after the uptake of Ce-ions into polarized macrophages and into unpolarized human monocytes and murine RAW 264.7 cells. These findings emphasize the need for evaluating ferritin biomineralization in studies that involve the internalization of nano objects, ranging from particles to viruses to biomolecules, to gain greater mechanistic insights into the overall immune responses to nano objects.

## 1. Introduction

With their unique physicochemical characteristics, NPs are extensively used to help revolutionize the biomedical, electronics, and energy sectors [1,2,3,4,5]. However, cellular nanoparticle (NP) exposure [6,7] has been linked to cytotoxicity [8,9,10,11,12]. Continuous streams of engineered NPs have been released into the pool of airborne particulates and further raise risk factors for nanoparticle-induced neurodegeneration and impaired cognitive function, among other diseases [13,14]. Macrophages, one of the most effective cell types in the body’s emergency response system, help clear NPs. Macrophages are leukocytes that act as scavenger cells that self-renew or develop from bone marrow-derived monocytes. They are part of a defense mechanism and regulate heterogeneous immune cells after NP exposure [15,16,17]. Although it is well established that NPs can cause immune system disruption, the mechanism(s) of the underlying inflammatory responses are complex. The discovery that NPs can perturb the polarization and reprogramming of macrophages and alter their immunological function is highly significant [18]. Macrophages can change (polarize) into different phenotypes to accomplish diverse roles under various physiological and pathological conditions and continuously adjust to the dynamic changes in the residing milieu [19]. Macrophage phenotypes play different roles in inflammation, immune regulation, proliferation, and cell metabolism. Their reprogramming can play an essential role in maintaining the steady state of the immune system [4,20]. Even after NP internalization and the ensuing production of cytokines that causes various immune reactions, macrophages retain their structural and functional flexibility [21,22,23]. By responding to different microenvironments, primary macrophages (M0) can polarize towards a M1 phenotype in settings of inflammation in response to invaders, such as bacteria, intracellular pathogens, or NPs, while M2 polarization occurs in response to Th2-type settings, including helminth infection and asthma, among others [24]. Many local environmental stimuli can modulate M0 macrophage activation [25]. Typical in vitro experiments utilize type II interferon (IFNγ) along with lipopolysaccharide (LPS) to polarize primary macrophages (M0) to M1 and IL-4/IL-13 to polarize to M2, mimicking the exposure of primary macrophages to T-lymphocyte cell subsets that occur during extrinsic macrophage activation in vivo [26,27]. The M1 phenotype produces high amounts of pro-inflammatory Th1-type cytokines, such as tumor necrosis factor α (*TNF-α*), a potent paracrine and endocrine mediator of inflammatory and immune functions, interleukins IL-6 and IL-18, and low amounts of IL-10 [26]. The M2 phenotype is characterized by production of high levels of anti-inflammatory IL-10 and low levels of IL-12 and IL-23 [28]. M1 cells are key effector cells in the resistance to intracellular pathogens and tumor growth [29,30], while M2 cells are associated with Th2-driven inflammation, immune regulation, promotion of tissue remodeling, and tumor progression [30]. It is critical to understand the fate of macrophages and to assess the biological matrices that can influence nanoparticle internalization, which can be accomplished with analytical imaging [31,32,33,34,35]. As an example, a greater uptake of silica NPs has been reported in M1 versus M2 macrophages, and this differential uptake provides a distinct repertoire of phagocytosis receptors in both cell types [34]. Nanoparticles affect macrophages and vice versa (a two-way cause and effect relationship). In receptor-mediated uptake, scavenger receptors are involved in the recognition and internalization of NPs [36]. Furthermore, the importance of a protein corona working in concert with scavenger receptors was recently shown to involve receptor-mediated uptake with cell-specific recognition of nanoparticles [37]. Learning how internalized NPs affect different macrophage phenotypes and control inflammatory diseases is expected to help gain insights into the regulatory effects of NPs on immune cells. This has emerged as a target for controlling inflammatory diseases.

Recent advances in analytical imaging, such as cryo-electron microscopy (Cryo-EM), coupled with electron energy-loss spectroscopy (EELS) and high-resolution scanning transmission electron microscopic (HRSTEM) analysis help gain deeper insights into nanoparticle–cell interactions and biotransformations and together allow tracing of chemical and structural fingerprints of the NPs inside the cytoplasm and in cellular organelles. After being phagocytosed, NPs may be bioprocessed, e.g., broken down, accompanied by intracellular synthesis of phases that are crystalline, amorphous, or represent a hybrid that incorporates organic molecules (derived from the protein corona of the phagocytosed NPs or cytoplasm) [37,38]. A distinct type of intracellular-formed NP are ferritin NPs (biomineralized iron), which frequently occur side-by-side with the phagocytosed NPs. This is critical because of the redox active nature of ferritin [38]. The two different subunits in the ferritin protein (shell around biomineralized iron), H and L, have different expressions driven by inflammatory stimuli; H-ferritin responds to inflammatory stimuli, while L- may act as an immunomodulatory molecule, displaying both pro-inflammatory and immunosuppressive functions. Since macrophages produce cytokines that cause serum ferritin secretion, it will be critically important to determine any variances in the formation of ferritin NPs in resident macrophages (M0) or M1- and M2-type cells after nanoparticle exposure and uptake. This needs to be accomplished for a range of biomedically and industrially important NPs.

For this study, we selected nanoceria (CeO_2_ NPs) because of their broad applications as a redox material and their acclaimed therapeutic properties that can provoke or attenuate an immune response. It has been demonstrated that CeO_2_ NPs can help control reactive oxygen species (ROS) inside cells, which has applications for the treatment of inflammatory diseases [39]. Specifically, CeO_2_ NPs can drive polarization to the M2 phenotype by controlling in situ-formed ROS, as shown in a stroke model [40]. Furthermore, when CeO_2_ NPs are decorated with additional catalytic promoter metals, a synergistic effect of beneficial oxygen generation and desired scavenging of ROS occurs during rheumatoid arthritis treatments [39]. CeO_2_ NPs that have not been calcined after solvothermal synthesis, which seems to include a broad range of nanoceria used in biomedical approaches, generally bioprocess after uptake, leading to the generation of cerium phosphate NPs inside macrophages [41,42]. Given significant differences in the uptake of some NPs by different macrophage phenotypes; the possibility that NPs induce a dissimilar response in different macrophage phenotypes, such as the formation of ferritin NPs; and the different effects on the immune system by NP polarization to M1 vs. M2 macrophages, it was deemed important to understand the interaction of CeO_2_ NPs and the different macrophage phenotypes. This current work compares differences in the intracellular fate of redox-active nanoceria to that of Ce-ion after internalization into primary macrophages (M0) and M1 and M2 cells, which has not been reported so far. Ce-ions were included because of the known release and redistribution of ions into the cytoplasm from dissolving (bioprocessed) CeO_2_ NPs [38], which is reminiscent of the biotransformation of gold NPs in fibroblasts [43]. We also compared the formation, intracellular sites, densities (number of particles), and possible role of ferritin NPs among phenotypes in response to nanoceria and Ce-ion internalization. Key questions address whether monocyte-formed (M0) compared to polarized M1 and M2 cells have marked differences in ferritin NPs’ accumulation and location in human and murine macrophages. The hypothesis that iron accumulation occurs as a response mechanism to the internalization of invader CeO_2_ NPs and Ce-ions in all phenotypes was examined.

## 2. Materials and Methods

### 2.1. Human Peripheral Blood Monocyte (M0) Source, Macrophage Differentiation, and Polarization to M1- and M2-like Cells

Human peripheral blood monocytes were obtained from de-identified human buffy coat bags and leukoreduction filters obtained from the Kentucky Blood Center. Flow through leukoreduction filters was reversed, and the cells collected. Further purification was performed by using Ficoll-Paque density gradient media and centrifugation, followed by additional monocyte magnetic purification using an EasySep Human Monocyte Isolation Kit (STEMCELL Technologies, Cambridge, MA, USA). Cells were incubated with an antibody cocktail for negative selection along with magnetic beads and placed in an EasySep magnet to remove unwanted cells and platelets. To differentiate them from macrophages, remaining cells (CD14^+^CD16^−^ monocytes) were cultured for six days at 1 × 10^6^ cells/mL in complete medium (RPMI 1640 High Glucose (Life Technologies, Carlsbad, CA, USA) with 50 ng/mL recombinant human macrophage colony stimulating factor (Peprotech, Rocky Hill, NJ, USA), 10% fetal bovine serum (Life Technologies), and penicillin/streptomycin (Life Technologies). Media was replaced every 2 days. Cells were washed and polarized with human cytokines to M1- and M2-like phenotypes with IFNγ (eBioscience, San Diego, CA, USA) 20 ng/mL and IL4/IL13 (PeproTech, Rocky Hill, NJ, USA) 20 ng/mL, respectively, for 24 h. LPS (strain *E. coli* 0111:B4, Invitrogen, Waltham, MA, USA) (20 ng/mL) was then added to each well to incubate for an additional 24 h. Cells were verified to have undergone macrophage differentiation by evaluating surface protein expression (CD14, CD11b, and CD163) and polarization through up-regulated surface expression of MHCII (M1) (anti-human MHC II antibody) (eBioscience) and CD206 (M2) (anti-human CD206 antibody) (eBioscience) evaluated by flow cytometry (Attune Acoustic Focusing Cytometer, Invitrogen). Results showed high expression of the M1 marker MHCII in 44% and 14% of cells polarized to M1 and M2, respectively. Conversely, high expression of the M2 marker CD206 was observed in 19% and 54% of the M1 and M2 polarized cells, respectively. After polarization treatments, the harvested M1- and M2-like cells were washed with PBS and exposed to CeO_2_ NPs or cerium ions

### 2.2. Murine Macrophage Polarization to M1 and M2 Cells and Polarization Verification

RAW 264.7 cells were obtained from the ATCC (TIB-71™, lot 62654190) and were seeded at 25,000 or 50,000 cells/well in 60 mm dishes in DMEM (Gibco (Waltham, MA USA), high glucose + glutamine + pyruvate) with 10% FBS (Gibco, endotoxin limit <5 EU/mL) in the absence of antibiotics. All studies were conducted with cells passaged no more than 13 times and polarized beginning approximately 18 h after seeding to M1- and M2-like phenotypes as conducted by others (24 h exposure to LPS from Escherichia coli O127:B8 (0.5 µg/mL) or IL-4 (20 ng/mL recombinant murine IL-4 (PeproTech, Cranbury, NJ, USA)) in complete or phosphate-free medium [44].

Several procedures were conducted to verify that the procedures to polarize the RAW cells to M1- and M2-like phenotypes were successful. RAW cells were grown in 24-well plates. After polarization, the medium was aspirated; the cells were scraped into 0.5 mL PBS; centrifuged to pellet the cells, which were resuspended in 0.1% Triton-X 100-PBS with the addition of a Roche protease inhibitor cocktail tablet; vortexed; centrifuged to remove non-lysed cells; and the supernatant collected and stored at −80 °C. An arginase assay was conducted [45] of non-polarized M1-like and M2-like cells. IL-4 increased arginase activity, but LPS did not [46]. Cell polarization was also assessed using the standard Seahorse XF Cell Mito Stress Test on a Seahorse XFe96 Analyzer (Agilent, Santa Clara, CA, USA). LPS exposure produced almost complete mitochondrial shutdown [46], consistent with other studies [47,48]. Cell polarization was further assessed by IL-1β release assay by Western blotting. IL-1β was decreased in M1-like cells [46], presumably due to decreased metabolism. Further evidence of LPS-exposed RAW cell polarization to the M1-like phenotype was shown by their greater size than unpolarized cells [46], as reported by others [49,50]. We used ImageJ (version IJ 1.46r) to quantify morphological features of the macrophages that showed the M1-like cells exhibit morphological differences, including increased size and enhanced intracellular vacuoles [46], as previously described [51,52].

### 2.3. Nanoparticle Synthesis

A solvothermal (hydrothermal) synthesis method was used to precipitate citrate-coated CeO_2_ NPs under autogenous pressure [53]. The NPs were dialyzed to ensure that no residual Ce-ions were adsorbed or physically trapped inside pores, as this would cause problems during the exposure of CeO_2_ NPs to macrophages by contributing unintended Ce-ion effects.

### 2.4. Cell Treatment Procedures

Three experiments were conducted with polarized human macrophages exposed to 10 μg/mL (as cerium) CeO_2_ NPs for 4.5 h. Nine experiments were conducted with polarized RAW cells exposed to 0 (PBS), 1, 3, or 10 μg/mL (as cerium) CeO_2_ NPs or 10 μg/mL Ce-ion for 6 or 24 h. To expose the cells, the medium was replaced with a phosphate-free medium (DMEM (or MP Biomedicals RPMI 1640 without phosphate and with 0.85 G/L sodium bicarbonate)) with 10% FBS. Complete DMEM medium has 0.915 mM phosphate. Nine percent FBS has ~8 µg/mL phosphorus (~0.25 mM phosphate). Total phosphate in DMEM + FBS was ~1.16 mM; in phosphate-free medium, ~0.25 mM. The CeO_2_ NPs and Ce-ion (Aldrich, Burlington, MA, USA, ICP/DCP standard solution) were introduced in 110 mM citric acid at pH 7.4. Cells were harvested after 6 or 24 h exposure, or the medium replaced with DMEM with 10% FBS and harvested up to 2 weeks later. For microscopy, they were washed with PBS and preserved in 2.5% PBS-buffered glutaraldehyde. Cells were processed for light microscopy and EM visualization and were not stained nor were heavy metals (e.g., osmium) added.

### 2.5. Analytical Imaging Techniques for NPs in Tissues

Analytical imaging of nanoparticles inside cells requires the use of high-voltage ~200 KeV applications. We utilized a JEOL 2100F STEM with a field emission gun at the Cincinnati NIOSH Electron Microscopy Lab that was equipped with an Oxford Aztec EDS (with a silicon drift detector) and a GATAN Tridiem EELS imaging filter. The crystal structure of tissue-trapped nanoparticles inside cellular matrices was analyzed using electron diffraction. Individual crystalline nanoparticles were analyzed at a lattice resolution of down to 0.15 nm using aberration-corrected STEM. HRSTEM is equipped with bright and dark field detectors and can analyze chemical information of translocated particles inside cells and detect metal ions in the cellular environment. Chemical analysis and mapping of nanoparticles helps determine if they react inside cells and undergo aging (chemical or physical breakdown/bioprocessing). A typical thin section embedded in plastic and mounted on formvar will not survive the high-energy electron beam unless the beam is spread, which depresses resolution. To solve these problems, we coated a standard mounted thin section on formvar with a thin evaporated carbon film or mounted the thin section on a carbon film grid, which stabilized the section under the high-energy beam, making them quite robust. No heavy metal staining was performed since this significantly increases the difficulty of locating NPs at the subcellular level. HRSTEM was operated using a high-angle annular dark field (HAADF) detector, which allows one to observe the ultrastructure in M1 versus M2 cells. It also makes it possible to monitor CeO_2_ NP biotransformation and bioprocessing, which may include degradation, dissolution, and reformation into secondary particles (i.e., CePO_4_). HRSTEM was used to image local cellular milieus adjacent to a nanoparticle at near-atomic resolution. The use of the STEM imaging mode allows ready use of a scanning probe as an analytical tool to detect and measure nanoparticle and tissue composition changes. Simultaneous application of energy dispersive spectroscopy (EDS) and EELS provides chemical information on the tissue environment as well as the physiochemical nature of translocated NPs. Specifically, EDS is used to measure compositional changes in point mode and in mapping mode, where it provides 2D composition maps. EELS analysis is used to determine valence changes in NPs in point mode and in line scan or mapping mode, where the valence changes are measured either across NPs (one dimensional) or imaged in two dimensions.

## 3. Results and Discussion

### 3.1. Nanoceria Physicochemical Properties

Synthesized CeO_2_-NPs (nanoceria) are citric acid (C_6_H_8_O_7_)-coated ~2–5 nm-sized crystallites with a cubic fluorite structure (Fm-3m), predominant (111) surfaces, and an average size of 4.2 nm, which combine into ~8–25 nm agglomerates (average size 15 nm) (Figure 1a–d). The CeO_2_ NPs have an average hydrodynamic radius of ~10.8 nm and a −40 mV surface charge at physiological pH. HRSTEM, electron diffraction, EELS analyses, and elemental mapping of the synthesized CeO_2_ NPs are summarized in Figure 1e–h and crystal details are provided in Appendix A. The measured lattice constant of 0.5412 from electron diffraction analysis indicates an expansion in the crystal structure (Appendix A). EELS shows an increased oxygen storage capacity (oxygen vacancies) (0.5%) for agglomerates that contain ~5 to 20 crystallites based on the Ce edges with characteristic elevated Ce M5 over Ce M4 signals (Appendix A). EELS analyses for the core and shell regions show an enrichment of Ce^3+^ at the particle surfaces (Figure 1g,h). This was confirmed with EELS of copious CeO_2_ NPs and corresponds well with structure investigations using neutron pair distribution function and Raman scatterings that indicate displacement of O_2_^−^ anions, as well as first-principle calculations, which show oxygen vacancies stabilize the crystallites [54]. We measured 2 nm-sized CeO_2_ NPs and show a lattice distortion/expansion of ~5%. EELS analysis suggests substantial incorporation of Ce^3+^, which is balanced by oxygen vacancies to give the crystallites their oxygen storage capacity (OSC) and significant redox activity (Appendix A) or radical scavenging potential. The measured OSC is greatest in the smallest crystallites, and our synthesized CeO_2_ NPs include particles below 2 nm, which contribute to the overall high OSC of ~7%. The locations of oxygen vacancies in a single 1.5 nm CeO_2_ crystallite are shown in (Appendix A). A high-defect structure in CeO_2_ NPs compared with bulk ceria has been used previously to explain fast dissolution after the cellular uptake of uncalcined CeO_2_ NPs [38]. Despite being uncalcined, the as-synthesized CeO_2_ NPs show crisp lattice images, but the crystallites are covered by an amorphous carbon-rich shell (Appendix A). We attribute the shell to citric acid-derived moieties or reaction products thereof.

In addition to citric acid, thin film water structures play critical roles and may deposit on the CeO_2_ (111) surfaces since the particles were suspended in aqueous media. Density functional theory (optPBE-vdW functional) was previously used to compute the stability window of CeO_2_ (111) surfaces covered with 0.5–2.0 mL of water and demonstrated that water adsorption layers depend on ceria’s complex defect structure [54]. The CeO_2_ NPs, therefore, may have both carboxylic acid and water adsorption layers.

### 3.2. Macrophage Polarization and Nanoceria Exposure

Macrophages obtained from human blood and murine RAW 264.7 cells were seeded and grown as M0 cells, which were then polarized to M1- and M2-like cells (Figure 1i,j). The phenotypic cells were then exposed to CeO_2_ NPs (10 µg/mL), Ce-ions (10 µg/mL), or vehicle (controls) for different exposure times to determine how exposure to and internalization into different macrophages affects the fate of CeO_2_ NPs and Ce-ions. Blood-derived monocytes (M0) were polarized to pro-inflammatory M1 using IFNγ and to anti-inflammatory-type M2 using IL4/IL13 and then exposed to CeO_2_ NP for 6 h or 24 h. RAW 264.7 cells were polarized to M1-type using LPS and M2-type with IL6, and then followed by exposure to CeO_2_ NP and Ce-ions (both for 6 h and 24 h).

### 3.3. Analytical Electron Microscopy of CeO_2_ NP Uptake and Biotransformation

How CeO_2_ NPs internalized into differently polarized human macrophages is detailed in Figure 2. HRSTEM analytical imaging was used to map M1- and M2-like cells to obtain the locations and cellular distribution patterns of internalized CeO_2_ NPs. Results revealed a differential internalization pattern and regions of uptake inside M1 versus M2 cells (Figure 2a–d). Specifically, it was shown that depending on the nature of the cells, the propensity for dynamic translocation of the CeO_2_ NPs into the interior of cells occurred in unique ways in M1 versus M2 (ΔE kinetic energy was needed to transport particles after entering the cytoplasm and disperse particles towards mitochondria, phagolysosomes, and other organelles).

A greater extent of localized uptake of CeO_2_ NPs by the M1 macrophages occurred immediately juxtaposed to the plasma membrane region. This postulates that the M1-specific NP-dispersion mechanism does not support great intracellular migration (Figure 2c). In contrast, in M2-like macrophages, under identical exposure conditions, the CeO_2_ NPs are found predominantly inside lysosomes or phagolysosomes throughout the cytoplasm, resulting in greater dispersion (Figure 2d). Monocyte-derived M0 cells exposed to CeO_2_ NPs analyzed to determine if there are differences between unpolarized and polarized cells regarding particle uptake showed that M0 cells have a very similar nanoparticle uptake mechanism as M1 polarized cells (Appendix A). Herein, we introduce the concept of intracellular *mechanical* forces that are unique to different phenotypes and that either hinder or promote particle transport inside cytoplasm and organelles. Recently, it was demonstrated that macrophages can sense external mechanical cues related to NPs with different surface properties, such as elasticity or stiffness, which allow cells to internalize different types of NPs at different rates, discriminating based on variations in elastic modulus [54]. Prior studies of non-polarized macrophages (M0) found CeO_2_ NPs adsorbed to the cell membrane and taken up by endocytosis, resulting in aggregates in vesicles and in the cytoplasm. Taken together, the results indicate a phenotype-specific and cytoplasm-controlled diffusion takes place after nanoparticle exposure. Nanoparticle recognition by cells via receptor-mediated entry [37] is then followed by phenotype-specific intracellular dispersion. Since the study used the same concentration (10 µg/mL) of redox-active CeO_2_-NPs and exposure times (6 h and 24 h) for M1 and M2 cells, the observed difference in CeO_2_ NP intracellular dispersion behavior, therefore, may be caused by differences in mechanical forces exerted by M1 versus M2 cells. Distinct NP-translocation tendencies, specifically the observed CeO_2_ NP buildup and sequestration at or near the membrane region in M1, further suggest that in activated M1 cells, the CeO_2_ NP distribution after endocytosis is transport limited. We can speculate that there is (a) a plasma membrane interaction with the entering CeO_2_ NPs that is different for M1 versus M2 phenotypes or (b) a chemical or physical change of the CeO_2_ NP surface, i.e., corona stripping (loss of surface carboxyl and hydroxyl functional groups) that alters the NP’s effective surface properties, which can change mobility towards M1 interior compartments when compared with the greater dispersion of CeO_2_ NPs into M2 cell interiors (Figure 2e,f).

To determine if the phenotypic cells interact with CeO_2_ NPs and cause either a chemical or structural transformation in the phagocytosed CeO_2_ NPs, analytical HRSTEM was performed (Figure 3). The results unequivocally show that human M1 (Figure 3a–k) and M2 (Figure 3l–r) cells both bioprocessed (chemically and structurally altered) the CeO_2_ NPs to Ce-phosphate nanoneedles.

Though it was previously shown that CeO_2_ NPs undergo biotransformation after uptake, i.e., inside plant roots and in alveolar cells [55,56], it was not known whether phenotypic cells would bioprocess NPs similarly or with completely different outcomes. HRSTEM analysis coupled with EDS provides information on the intracellular location, size, morphology, crystallinity, density, composition, and redox state of the CeO_2_ NPs, which allowed an assessment of the degree of bioprocessing that occurred in M1 versus M2 cells. Overall, it was determined that a fraction of CeO_2_ crystallites and agglomerates in both human M1 (Figure 3a) and M2 cells (Figure 3l) endure without chemical and structural change when compared with the as-synthesized materials. However, many partially dissolved CeO_2_ NPs, some as small as 2 nm, occur in both M1 (Figure 3b–d) and M2 cells (Figure 3m–o). This suggests intracellular dissolution must have occurred, which releases Ce-ions into the cytoplasm, as has been proposed in plants [57]. The partially dissolving CeO_2_ nanocrystals in both M1 (Figure 3a–d) and M2 (Figure 3l–o) cells act as nucleation sites for the intracellular nucleation and growth of Ce-phosphate, which forms crystalline nanoneedles. High-angle annular dark field (HAADF) and EDS mapping of the nanoneedles identify the presence of Ce, P, and O (Figure 3e–g,p–r). Electron diffraction patterns for representative CeO_2_ NPs (Figure 3i) and Ce-phosphate nanoneedles (Figure 3j) were analyzed in TEM mode. The Ce-phosphate nanoneedle diffraction pattern was compared with the Materials Project [55] database and identified in both M1 and M2 cells as CePO_4_ with characteristic hexagonal phase-containing Ce^3+^ (Figure 3k). Although CeO_2_ NPs are typically seen juxtaposed to the Ce-phosphate nanoneedles, we also find copious needles without precursor CeO_2_ NPs in both M1 and M2 phenotypes. This points to two potential formation mechanisms of CePO_4_, including epitaxial growth on CeO_2_ facets and nucleation and growth. Which cellular constraints exactly control the formation of Ce-phosphate nanoneedles and whether they form on CeO_2_ surfaces or are free-formed cannot be resolved with this study, but we speculate it depends on both the availability of Ce-ions from the breakdown of CeO_2_ NPs and accessibility of phosphate ions within the M1 and M2 cells. The Ce-phosphate nanoneedles in M1 and M2 cells were comparable in size and had similar aspect ratios, and the same results were obtained for human monocytes (M0; Appendix A). The chemical transformation of CeO_2_ NPs to CePO_4_ nanoneedles (bioprocessing) is the same in all phenotypes (M0, M1, and M2), but particle dispersion (intracellular transport mechanisms) differs for the human M1- versus M2-like cells.

### 3.4. Nanoceria Cellular Distribution

The differential uptake of CeO_2_ NPs by RAW 264.7 M1- and M2-polarized cells was observed after 6 and 24 h exposure (Figure 4). Results indicate that the murine M1-type cells concentrated the CeO_2_ NPs after uptake close to the plasma membrane surface region (Figure 4a), analogous to human M1 cells. Murine M1 cells also show intracellular uptake and dispersion of CeO_2_ NPs to be diffusion-limited. In contrast, the M2-like cells internalized the CeO_2_ NPs throughout the cell interior and into lysosomes and phagolysosomes (Figure 4h), as was the case for human M2 cells. Both murine M1 (Figure 4b–d) and M2 (Figure 4i–k) cells bioprocessed the internalized CeO_2_ NPs (6 h and 24 h exposure) to Ce-phosphate nanoneedles that displayed similar size and aspect ratios as those found in human M1 and M2 cells, suggesting the same underlying mechanism(s) must be operating in human and murine phenotypes. Elemental maps for Ce, P, and O (Figure 4l–n) confirm the location of Ce-phosphate nanoneedles juxtoposed to bioprocessed CeO_2_ NPs. EELS analyses were compared for the CeO_2_ NPs and Ce-phosphate nanoneedles in both M1 (Figure 4o) and M2 (Figure 4p) cells, showing that the composition and cerium oxygen edges are the same and that bioprocessing is the same in both phenotypes.

Exposure of murine M1 and M2 cells to Ce-ions (10 µg/mL; 6 h and 24 h) demonstrated that the Ce-ions gained access to the entire cytoplasm and nucleate and grew Ce-phosphate rod-like NPs instead of needles throughout M1 and M2 cytoplasm, specifically inside copious phagolysosomes and, to a lesser degree, inside mitochondria (Figure 5). Similar effects occur in unpolarized RAW 264.7 cells after Ce-ion exposure (Appendix A). HRSTEM, TEM, and EDS mapping and electron diffraction revealed no measurable difference in Ce-ion uptake and subsequent precipitation as CePO_4_ nanorods in either murine M1 and M2 cells (Figure 5a–n). The CePO_4_ nanoneedles involve individually aligned ~1–2 nm crystalline nanorods that assembled to form dendrite-like superstructures in both murine M1 (Figure 5f) and M2 cells (Figure 5m).

Importantly, this demonstrates that Ce-ion uptake followed by intracellular ionic transport throughout the cytoplasm is not macrophage-phenotype controlled. The morphology of the CePO_4_ nanorods that formed after Ce-ion exposure (Figure 5f,m) is different from the more elongated and narrower Ce-phosphate nanoneedles that formed after the CeO_2_ NP dissolution and bioprocessing mechanism (Figure 4c,j). The composition (CePO_4_) and crystallinity (hexagonal phase) are the same for rods and needles (Figure 4 and Figure 5). The information that was gained with analytical imaging helped (a) evaluate the bioprocessing of nanoparticles and ions in different macrophages, (b) detect and compare subcellular diffusion phenomena for nanoparticles and ions, and (c) measure elemental and structural details of translocated, bio-transformed, or intracellularly precipitated NPs in human and murine macrophage phenotypes. Taken together, our findings suggest that NPs that enter M1 and M2 cells are affected by phenotype-specific controls that restrict migration and intracellular dispersion of particles differently in M1 versus M2 cells, while Ce-ions translocate and migrate without barriers in M1 and M2 cells and unpolarized cells (human monocyte-derived M0 and RAW 264.7 cells). The type of mechanical forces that act on the NPs and keep particle translocation limited to certain subcellular regions in M1 versus M2 cells could not be documented here since it is not possible to measure the energies required for NP movement in situ of the phenotypic cells. This should be analyzed using environmental TEM applications with time-processed video imaging in future studies. The internalization routes for M1 versus M2 phenotypes have a determining effect on the fate, post-intracellular localization, and dispersion of the invader CeO_2_ NPs but not for Ce-ions [56].

Translocation and intracellular diffusion energies for nanoparticles depend on local environmental conditions and on surface properties, such as corona, ligand accumulation, and surface functionalization, which would affect the particle’s free movement in phenotypic cells. As suggested for the translocation of TiO_2_ NPs through gut epithelium [58], particles, upon crossing, can alter the paracellular permeability of the colon epithelia, which affects transport and migration and leads to integrity alteration. Subcellular modulation of nanoparticle–cell interactions and uptake through the plasma membrane are formidable challenges [59]. Surface chemistry and corona formational processes can effectively reduce NP surface energy [60], affect surface charge, and modulate the transport phenomena of NPs in cells. It was demonstrated that macrophages could sense external mechanical cues related to NPs with different surface properties, such as elasticity or stiffness, which allows cells to internalize different types of NPs either quickly or slowly based on variations in the elastic modulus of the invader particles [54]. The different translocation and dispersion of CeO_2_ NPs in M1 versus M2 phenotypes point to a variation in particle surface energy (ΔE), which is at least in part controlled by corona, composition, and charge. This can affect CeO_2_ NPs during and/or after entering subcellular space. Cells can sense external mechanical cues, such as hardness, stiffness, and elastic modulus exerted by invading NPs, to control the degree of cell–NP interactions, including cellular binding and endocytosis rate [54]. The elasticity of different cell-invading particles was shown to be a controlling factor in how macrophages respond to NP uptake and how effectively NPs can enter cells. Soft and rigid NPs that otherwise have the same composition and size were shown to have different cell internalization behavior [54]. In the case of CeO_2_ NPs, the elastic modulus varies depending on the level of oxygen vacancy concentration, specifically, the greater the presence of vacancies, the lower the strength of the chemical bonds in the CeO_2_ NP lattice, which decreases their elastic modulus [61]. Oxygen vacancies within the surface of CeO_2_ NPs must be charge-balanced by Ce^3+^, while the core contains mostly Ce^4+^ (Appendix A). Oxygen vacancies give CeO_2_ NPs their characteristic free-radical scavenging activity [62] and elastic modulus and, thereby, affect the NP–cell interaction activity. The interaction of an oxygen-deficient CeO_2_ NP with M1 or M2 cells may, in part, determine uptake, cell binding, and bioprocessing effects.

HRSTEM investigations revealed a phenotype-dependent internalization for CeO_2_ NPs by tracking the location of the particles at the cell membrane region and through the cell’s interior. Location-specific identification of the internalized CeO_2_ NPs in M1 versus M2 cells demonstrated a notable difference in the internalization kinetics of CeO_2_ NPs that can only be explained as being guided by M1 and M2-specific signaling pathways. Since CeO_2_ NPs are trafficked more completely throughout the interior of M2 cells and are more marginalized to the membrane region in M1 cells, this suggests that phenotype-specific uptake involves phenotype-unique transport kinetics. Therefore, our findings hold significance for NP interactions in tissues that solicit macrophage polarization.

### 3.5. Nanoceria and Cerium Ion Effects on Macrophage Polarization

Light microscopy of murine M1- and M2-like cells after uptake of CeO_2_ NPs or Ce-ions indicated that there are measurable differences in cell nucleus shape and size compared with unexposed cells (Appendix A). Analyses of the nucleus size (using ImageJ (version IJ 1.46r)) before exposure began showed M1-like cells to have a larger nucleus than either M0 or M2-like cells, which have a similar size (Table 1). However, when comparing the nuclear sizes of M0- and M1-like cells after treatment with CeO_2_ NPs, it became apparent that the ceria–cell interactions after uptake had led to a reduction in the nucleus size of pro-inflammatory M1-like cells, effectively making them, at least as far as size is concerned, more like M0 and more like pro-healing/anti-inflammatory M2 cell nuclei (Table 1). Exposure to Ce-ions did not have this effect after either 6 h or 24 h exposure. We presume this is causally related to the lack of redox activity of Ce-ions and reduced redox activity of Ce-phosphate nanoneedles compared with the study’s synthesized redox-active CeO_2_ NPs (Appendix A). Bioprocessing of CeO_2_ NPs in phenotypic cells (human and murine) was shown to be the same in M1- and M2-type cells, yet the uptake of CeO_2_ NPs seems to only affect M1 by effectively reducing the nucleus size of M1 (S6) and triggering a structural resizing of the M1-nucleus to become more M2-like in morphology, as illustrated schematically in Figure 6. Cell circularity was 0.89, 0.81, and 0.89 for M0, M1, and M2 cells at 6 h and 0.89, 0.79, and 0.88 for M0, M1, and M2 cells at 24 h, respectively (Appendix A). The M1 cell area was significantly greater than M0 or M2 cells at 24 h, 103, 41, and 38 μm^2^, respectively. Taken together, CeO_2_ NPs can be involved in the reprogramming or shifting of the pro-inflammatory type M1 away from M1 likeness or towards more pro-healing M2 (for potential therapeutic benefits). This nucleus size change or restructuring is illustrated using ZONES I, II, and III in Figure 6. A similar response to CeO_2_ NP exposure was seen in CD4^+^ and microglial cells [63,64] and involves a restructuring of the nucleus, which may have some implications for potential gene therapy applications.

### 3.6. Ferritin Biomineralization after CeO_2_ NP and Ce-Ion Exposure

As mentioned in the introduction, no previous data are available regarding phenotype-related ferritin NP formation inside macrophages. Ferritin NPs have been linked in our previous works to the sites of invader particles and metal ions at subcellular levels [38,65]. High blood serum ferritin levels typically accompany inflammation and disease, whereas serum ferritin protein has little or no iron content [66]. On the other hand, ferritin NPs are biomineralized iron oxyhydroxide (FeHO_2_) that form ~5–10 nm NPs inside the cage of the iron storage protein [67]. STEM–HAADF analysis in the current study was able to identify copious ferritin NPs in both M1 and M2 phenotypes (human and murine cells) after exposure to either CeO_2_ NPs or Ce-ions (Figure 3, Figure 4 and Figure 5). Likewise, ferritin NP accumulation was observed in the unpolarized human (M0) cells that were exposed to CeO_2_ NPs and compared to the ferritins in M2 cells exposed in the same way (Appendix A) and also with unpolarized RAW 264.7 cells (Appendix A). In all phenotypes, swarms of ferritin NPs formed in the immediate vicinity of the translocated CeO_2_ NPs (Figure 5m and Figure 6) and alongside Ce-phosphate nanorods that precipitated in the cells after Ce-ion uptake (Figure 5f,m). This suggests that nanoparticle exposure, together with metal ion uptake, leads to an upregulation of iron in all phenotypic cells (M0, M1, and M2). Ferritin NPs are recognized as solitary dark (TEM) or bright (STEM) spots in the cytoplasm, and only the mineralized portion is seen because the protein shell is electron-poor and has minimal contrast against the cytoplasm. The ferroxidase in the protein shell catalytically controls iron storage as part of the iron metabolism and enables the ingress of Fe^2+^ ions into the protein cage (apoferritin), which then become enzymatically oxidized to form the iron core [68]. The core has up to 5000 iron atoms [68]. Notably, for each Fe^2+^ oxidation to Fe^3+^, an electron is released that can become available in free-radical scavenging or antioxidant reactions [52]. To analyze individual ferritin NPs inside the macrophage’s cytoplasm, Z-contrast high-angle annular dark-field (HAADF) STEM mode is used. Next, we focused on a M1-type cell that was exposed (24 h) to CeO_2_ NPs and contained bioprocessed Ce-phosphate nanoneedles with abundant ferritin NPs, identified with arrows in Figure 7a,b. At higher magnification in STEM mode, an EDS map of ferritin NPs together with Ce-phosphate nanoneedles shows the close special arrangement in the cytoplasm of M1 (Figure 7c). The corresponding EDS spectrum for the same area of the EDS map is shown in Figure 7d. Ferritin NPs can also be present in swarms in regions where no CeO_2_ NPs or phosphate needles are seen (Figure 7e). An EDS line scan (Figure 7f) was obtained across Ce-phosphate nanoneedles in the M1 cell to observe if iron concentration was increased at the exact location of the Ce-nanoneedles (Figure 7g,h).

Profiles of Ce, P, and Fe counts along the trace line indicate that the Ce and P signals are correlated, but the Fe signal is not, suggesting that ferritin NPs form in the vicinity of the invader NPs but not at the exact same location. EELS analysis can be used to measure an individual ferritin NP in the cytoplasm to gain information on the oxidation state for iron. We analyzed a 7 nm ferritin NP located inside an M1 cell and juxtaposed to internalized CeO_2_ NPs and identified the FeL_2,3_ spectrum, which showed a large FeL_3_ (709.1 eV) and small FeL_2_ (724.1 eV) peak characteristic of predominantly Fe^3+^ in the iron core (Figure 7i). EELS spectra for ferritin NPs in M1 and M2 cells after uptake of CeO_2_ NP and ionic Ce are compared in Appendix A. Results indicate that there is an exceedingly small variation of only 1.5 eV in the shift between FeL_2_ and FeL_3_ peaks for ferritin NPs analyzed in both the cytoplasm and lysosomes of M1- and M2-type cells. This suggests that ferritin NPs in both phenotypic cells have predominantly Fe^3+^ valence. Ferritin NPs analyzed with aberration-corrected STEM (HAADF bright field detector) appear dispersed throughout M2 cells and concentrated in lysosomes but are more localized towards the membrane in M1-type cells where CeO_2_ NPs gather, shown in the schematic of Figure 6. Ferritin NPs in M1 and M2 cells have similar morphology and size but different distribution in lysosomal versus cytoplasmic regions (Figure 6). Particularly, the close locality of ferritins with CeO_2_ NPs and bioprocessed CePO_4_ (needles/rods) suggests iron accumulation as a response mechanism to the internalization of invader particles. This indicates that ferritin NPs shadow the areas where more intruder NPs occur and, therefore, may be correlated with more inflammation, which has been shown in areas of elevated inflammatory responses [61,62]. In the case of Ce-ion exposure, both M1- and M2-type cells contain copious ferritin NPs in the cytoplasm and packed inside lysosomes with no recognizable difference among the phenotypes, which was confirmed with EELS data obtained for copious ferritins and shown for two individual ferritin in Appendix A. It is clear from the STEM analyses that invader CeO_2_ NP and ionic Ce uptake by M1 versus M2 cells increases or activates iron influx and uptake, but M2-type cells have more ferritin NPs in lysosomes compared to M1 cells and have a greater density distribution of lysosomes (schematic in Figure 6). To our best knowledge, this is the first report of differences in ferritin NP location and density in M1- versus M2-type cells after NP or metal ion exposure.

There is also a marked difference in human and murine macrophages in that chiefly the M1-type cells have accumulations of crystalline magnetite (Fe_3_O_4_ NPs). These are bigger in crystal domain size than ferritin NPs and occur in clusters, which makes them easily detectable in STEM (Figure 7j). Unlike ferritin NPs, Fe_3_O_4_ NPs contain both Fe^3+^ and Fe^2+^ in octahedral and tetrahedral sites, respectively. Electron diffraction analysis confirms that the Fe_3_O_4_ NPs have the typical spinel structure with predominantly (111), (220), and (311) faces (Figure 7n). A comparison of the oxidation states of three individual ferritin NPs and three Fe_3_O_4_ NPs in M1-type cells after exposure to CeO_2_ NPs shows there is a very narrow shift in the FeL_3_ peak in ferritin NP but no shift at all in the Fe_3_O_4_ NPs (Figure 7k). This suggests that ferritin NPs can contain minute amounts of Fe^2+^, which could become the seeds for Fe_3_O_4_ NPs formation in M1-type cells. It has been suggested previously that the breakdown of the ferritin cage can lead to the release of biomineralized iron to serve as a precursor for the synthesis of biogenic magnetite, which has been found in chronic diseases and was described in Alzheimer’s brain tissue [69]. It is thus plausible that the formational mechanism of biogenic magnetite (Fe_3_O_4_ NPs) depends on macrophage availability and ferritin NP accumulation density since greater ferritin NP availability would be needed to form larger Fe_3_O_4_ NPs in cells at sites of acute or chronic injury. Taken together, our results show that ferritin NPs first infiltrate cytoplasm and lysosomes in M1- and M2-type cells after CeO_2_ NP exposure by (i) precipitating oxidized iron inside the ferritin cage and (ii) some ferritin undergoes reduction so that both Fe^3+^ and Fe^2+^ are sequestered in a crystal spinel (Fe_3_O_4_ NP) with magnetic properties that vary from the initial ferritin NP.

## 4. Conclusions and Further Applications

Our results demonstrate distinct pathways for the internalization of CeO_2_ NPs into macrophages with different polarization status. We conclude that macrophage polarization controls the cellular distribution of internalized CeO_2_ NPs in human and murine M1 and M2 cells with distinct translocation, but with identical biotransformation and bioprocessing of the CeO_2_ NPs to Ce-phosphate nanoneedles. Our findings provide insights on the important role that polarization plays in sequestering NPs to certain cell compartments after initial uptake with M1 cells having more buildup at the cell membrane while M2 cells disperse nanoparticles throughout the entire cytoplasm. Bioprocessing was shown to be the same in M0, M1, and M2 cells after exposure to CeO_2_ NPs, and was demonstrated using high-resolution analytical imaging. Cerium metal ion uptake by M1 and M2 resulted in the same translocation and bioprocessing throughout cytoplasm for all cells. Macrophage polarization does not have any influence on ferritin formation and co-localization following CeO_2_ NP uptake or Ce-ion internalization into the phenotypic cells. Ferritin effects are also the same for blood-derived monocytes (M0) and unpolarized RAW 264.7 cells after CeO_2_ NP uptake. Physiologically relevant data on iron status and ferritin NP distribution after nanoparticle and metal ion uptake by different phenotypes results in the same biotransformation and bioprocessing in all macrophages. However, endogenous Fe_3_O_4_ NPs play a unique role in M1- versus M2-type cells after CeO_2_ NP internalization, with Fe_3_O_4_ NPs being more available in the inflammatory M1 type. The observation of ferritin buildup after nanoparticle or ion uptake in all macrophages may be part of the initiation stage of immune activation caused by cell invaders. The significance of high ferritin accumulation in the vicinity of internalized CeO_2_ NPs in both M1 and M2 macrophages and in unpolarized cells suggests that inflammatory cytokine release triggered by invader particles may activate endogenous iron formation, biomineralization, and accumulation in all phenotypes. The nanoscale analyses utilized here can be applied broadly to other toxicological outcomes with different invader NPs or metal ions, and future work should also include the internalization of viruses, e.g., COVID-19 [70,71,72,73,74]. The extraordinary ferritin densities observed here after ceria nanoparticle exposure and uptake inside all macrophage phenotypes may also be a contributing factor in the high iron concentration that is analyzed with magnetic resonance imaging (MRI) assessments of organs with high iron status. Previously, high iron status was utilized to gain insights into pre-symptomatic organ dysfunction [62,63,65,66]. Therefore, future work on nanoparticle-induced inflammation, toxicity, and the related ferritin buildup should also include advanced MRI imaging of tissues with ferritin-loaded macrophages to better understand how iron overload is associated with disease formation where inflammation-induced ferritin response are key.

## Figures and Tables

**Figure 1 nanomaterials-13-02298-f001:**
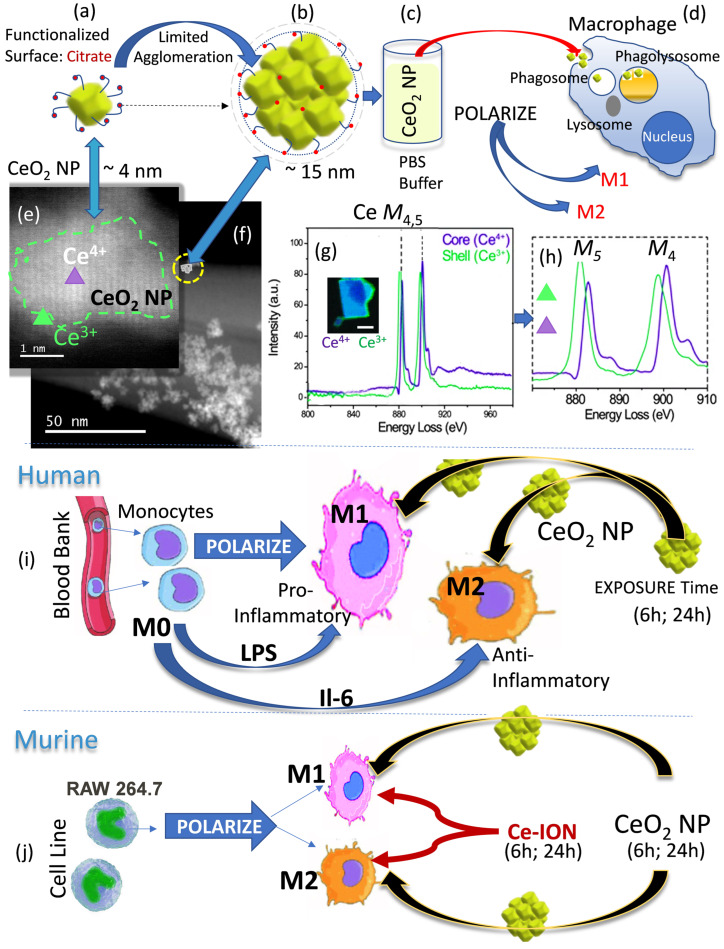
Nanoceria synthesis and exposure to macrophage phenotypes. (**a**,**b**) show a CeO_2_ NP with citrate surface coating as a single crystal and agglomerates; (**c**) is a schematic showing dispersion of CeO_2_ NPs in PBS buffer; (**d**) is a schematic of a polarized macrophage cell exposed to CeO_2_ NPs and Ce-ions; (**e**,**f**) are STEM images showing single domain and agglomerates of CeO_2_ NPs, respectively. The green dotted line and triangle in (**e**) mark the outer and inner regions of the CeO_2_ NP and location of EELS analyses. (**g**,**h**) are EELS analyses with the edges for Ce M_5,4_ for core and shell regions of the CeO_2_ NP single crystal shown in (**e**). (**i**) is a schematic illustration of the polarization of human blood-derived monocytes (M0) to M1 (IFNg; pro-inflammatory) and M2 (IL4/IL13; anti-inflammatory)-type cells and exposure of CeO_2_ NP for 6 or 24 h. (**j**) is a schematic illustration of the murine cell line (RAW 264.7) polarization to M1 (LPS) and M2 (IL-4)-type cells with exposure to CeO_2_ NP and Ce ions.

**Figure 2 nanomaterials-13-02298-f002:**
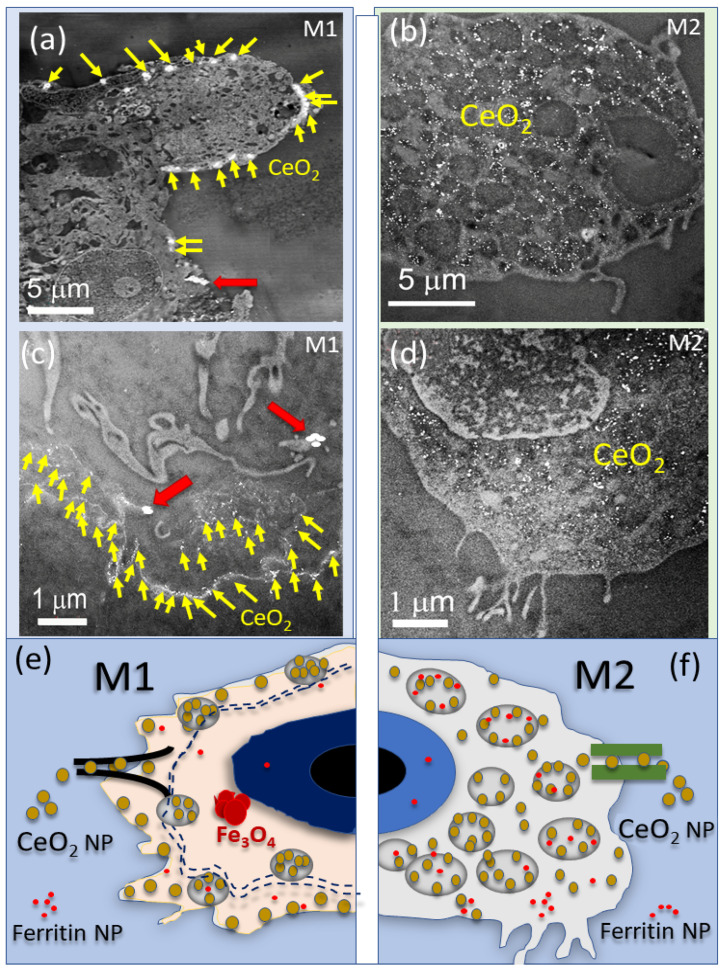
This figure illustrates STEM images of polarized macrophages after CeO_2_ NP exposure for 24 h and schematics of CeO_2_ NP internalization. (**a**) STEM image shows a M1 cell with CeO_2_ NPs as white particles with locations marked by yellow arrows, and (**c**) shows a magnified region. The red arrows mark the location of iron oxide (Fe_3_O_4_) nanoparticles. (**b**) STEM image of a M2 cell with internalized CeO_2_ NPs (white particles), and (**d**) shows a magnified region. (**e**,**f**) are schematics of the internalization of CeO_2_ NPs in M1 and M2 cells, respectively. The dotted lines in € mark the region at the M1 cell wall where CeO_2_ NPs and ferritin NPs are concentrated. Fe_3_O_4_ NP are present in M1. (**f**) is a schematic of CeO_2_ NP uptake into M2 cells with high dispersion into lysosomes and ferritin NP dispersion but no Fe_3_O_4_.

**Figure 3 nanomaterials-13-02298-f003:**
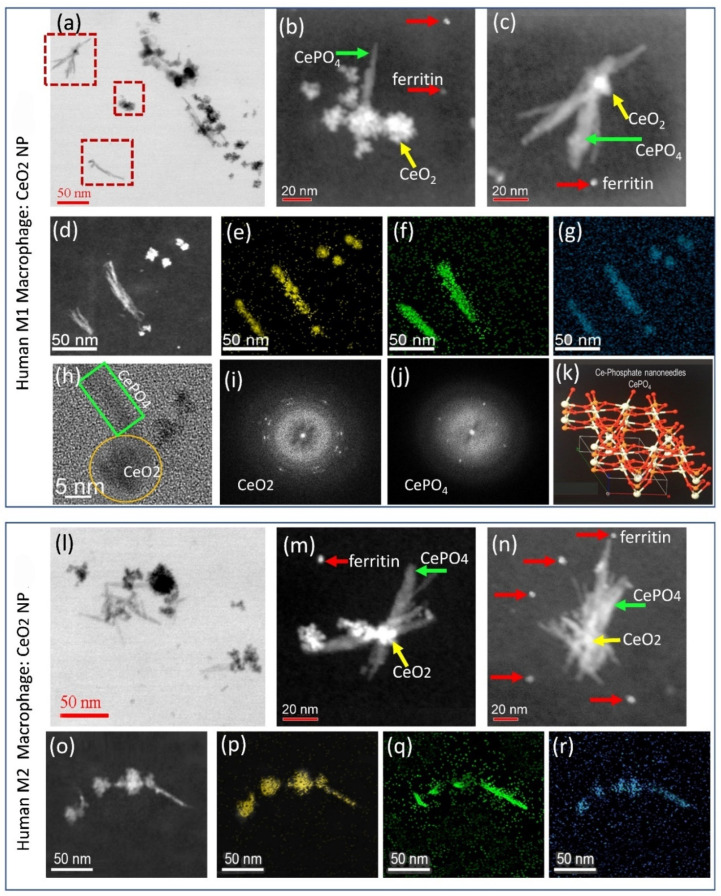
TEM and STEM images of CeO_2_ NPs in human macrophages after 24 h exposure. (**a**–**j**) illustrates M1 cells. (**l**–**r**) illustrates M2 cells. (**a**) TEM image of a human M1 macrophage shows CeO_2_ NPs as black particles and, in gray, the Ce-phosphate nanoneedles that grew from the surface of the CeO_2_ NPs or grew unattached. (**b**–**d**) are STEM images of magnified regions in the squares in (**a**) and show the presence of ferritin NPs. EDS maps for Ce, P, and O, corresponding to particles in (**d**), are shown in (**e**–**g**). (**h**) is a HRTEM image showing a CeO_2_ NP and CePO_4_ nanoneedle with corresponding electron diffraction in (**i**,**j**). The electron diffraction from the CePO_4_ nanoneedle corresponds to a crystal lattice of CePO_4_ from the Materials Project database (**k**). (**l**) TEM image of a murine M2 macrophage shows CeO_2_ NPs as black particles and CePO_4_ nanoneedles in gray. (**m**–**o**) STEM images showing CeO_2_ NPs and CePO_4_ nanoneedles with the same morphologies when compared with human M1 macrophages. Ferritin NPs are shown in (**m**,**n**). (**p**–**r**) EDS maps for Ce, P, and O.

**Figure 4 nanomaterials-13-02298-f004:**
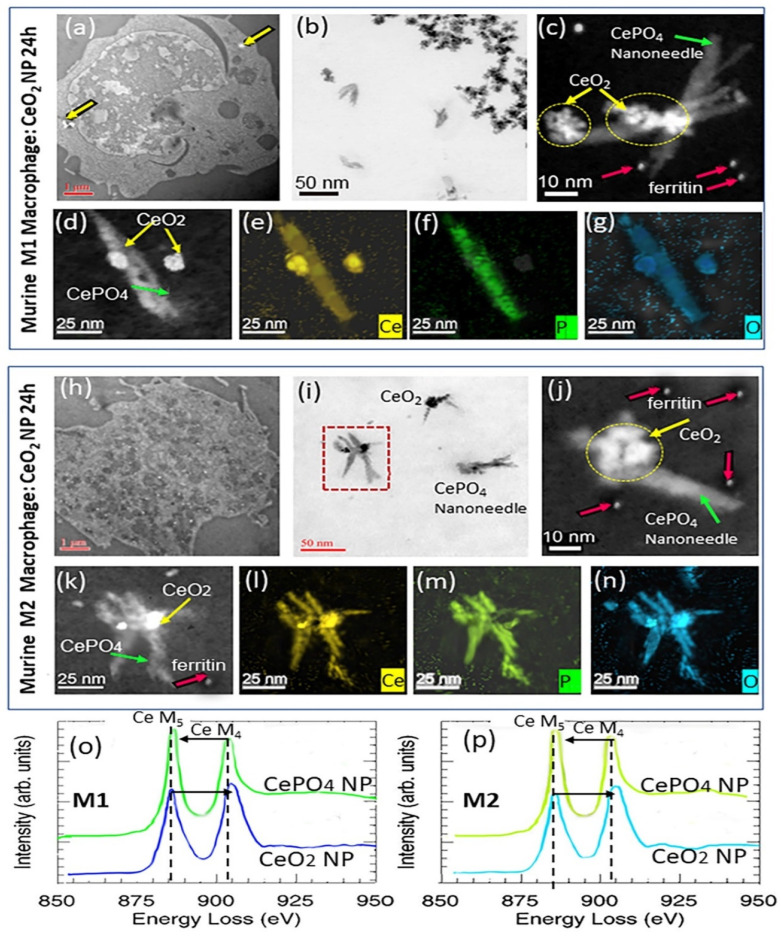
TEM and STEM images of CeO_2_ NPs in murine macrophages after 24 h exposure. (**a**–**d**) illustrate M1 cells; (**h**–**k**) illustrate M2 cells. (**a**) STEM of M1-like cell with CeO_2_ NPs close to the cell membrane region marked by arrows. (**b**) TEM shows a magnified region from (**a**) with copious CeO_2_ NPs and CePO_4_ nanoneedles that grew attached to CeO_2_ surfaces and free. (**c**,**d**) are STEM images of a magnified region in (**b**). (**e**–**g**) EDS maps for Ce, P, and O. (**h**) is a STEM image of a M2-like cell with dispersed CeO_2_ NPs throughout the interior of the cell. (**i**) TEM image of a magnified region from (**h**) showing CeO_2_ NPs and CePO_4_ nanoneedles with the same morphologies as in M1-type cells. (**j**,**k**) are STEM images of a magnified region in (**h**) with ferritin NPs present around CeO_2_ NPs and CePO_4_. (**l**–**n**) are EDS maps for Ce, P, and O of crystals in (**k**). EELS analyses with Ce M_5_ and M_4_ edges for a CeO_2_ NP and CePO_4_ NP in M1 (**o**) and M2 (**p**) cells, respectively. Ferritin NPs are marked with red arrows in (**c**,**j**,**k**).

**Figure 5 nanomaterials-13-02298-f005:**
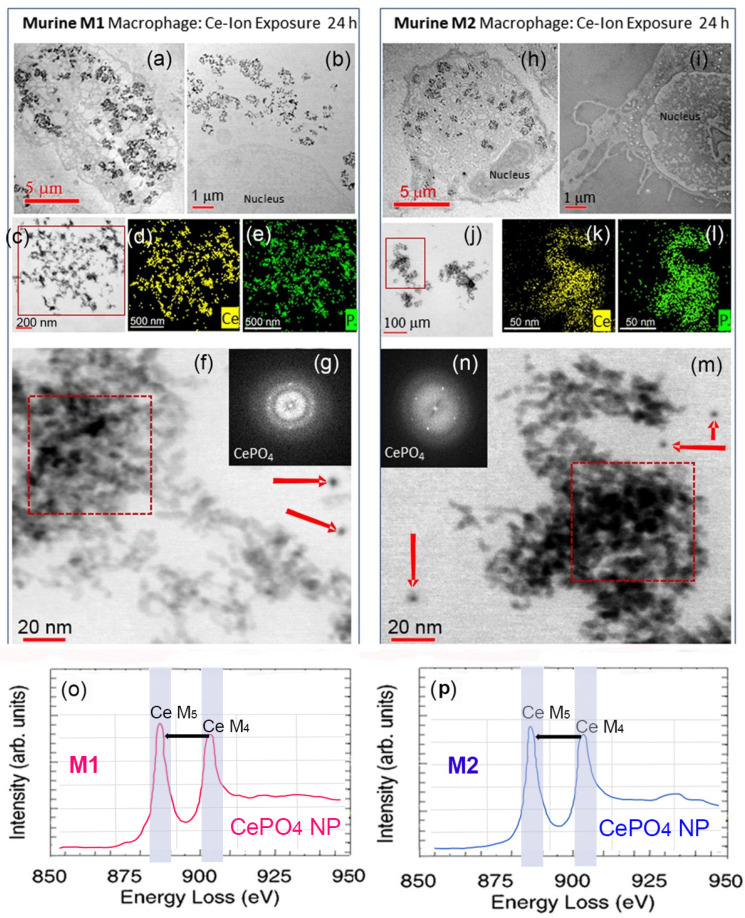
Murine M1 (**left column**) and M2 (**right column**) cells after Ce-ion exposure for 24 h. (**a**–**c**,**f**) are TEM images of nanoparticles formed in M1 and (**d**,**e**) are EDS maps for Ce and P of particles in (**c**). (**g**) shows electron diffraction of particles in (**f**), which correspond to CePO_4_. (**o**) EELS analysis of the particles in the red square in (**f**) with Ce M_5_ and M_4_ edges marked. TEM in (**h**) and STEM in (**i**) show murine M2 cells with precipitated nanoparticles. (**j**,**m**) are magnified regions from (**h**) with corresponding EDS maps (**k**,**l**) for Ce and P from the red square in (**j**). Electron diffraction of nanoparticles in the red square in (**m**) corresponding to CePO_4_ and EDS elemental distribution is shown in (**n**). EELS analysis in (**p**) of particles in (**m**) with Ce M_5_ and M_4_ edges shown. Red arrows in (**f**) and (**m**) mark ferritin NPs.

**Figure 6 nanomaterials-13-02298-f006:**
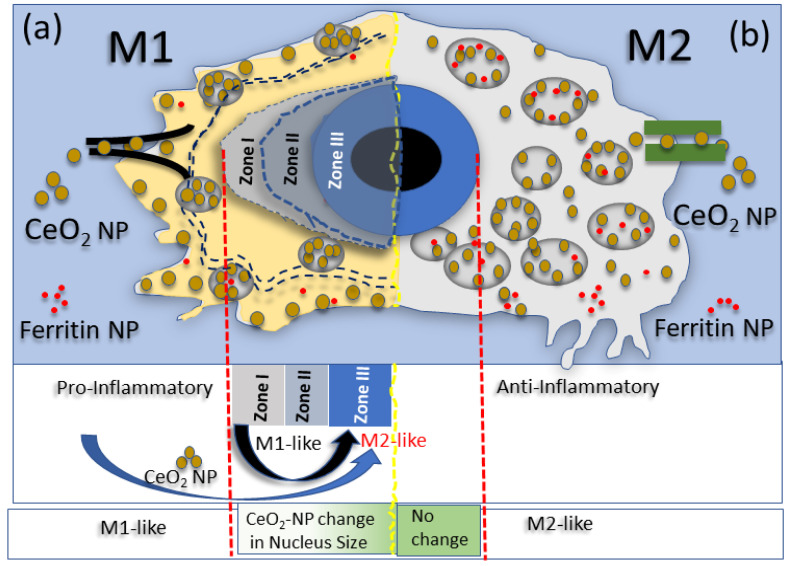
Schematic of M1 pro-inflammatory (**a**) and M2 anti-inflammatory (**b**) cells after exposure to CeO_2_ NPs with a characteristic distribution of CeO_2_ NPs at the cell membrane in M1 and throughout entire M2 cells in abundant lysosomes. This figure does not show the bioprocessing of CeO_2_ NPs to CePO_4_ NPs. Only M1-type cells undergo, after CeO_2_ NP exposure, a contraction of the nucleus size from Zone I to a smaller Zone II (intermediate) and Zone III, which is the same size as M2-type cells, indicated with the back and blue arrows in (**a**). Ferritin NPs are shown next to CeO_2_ NPs in M1 closer to the cell membrane and in M2 throughout the cell.

**Figure 7 nanomaterials-13-02298-f007:**
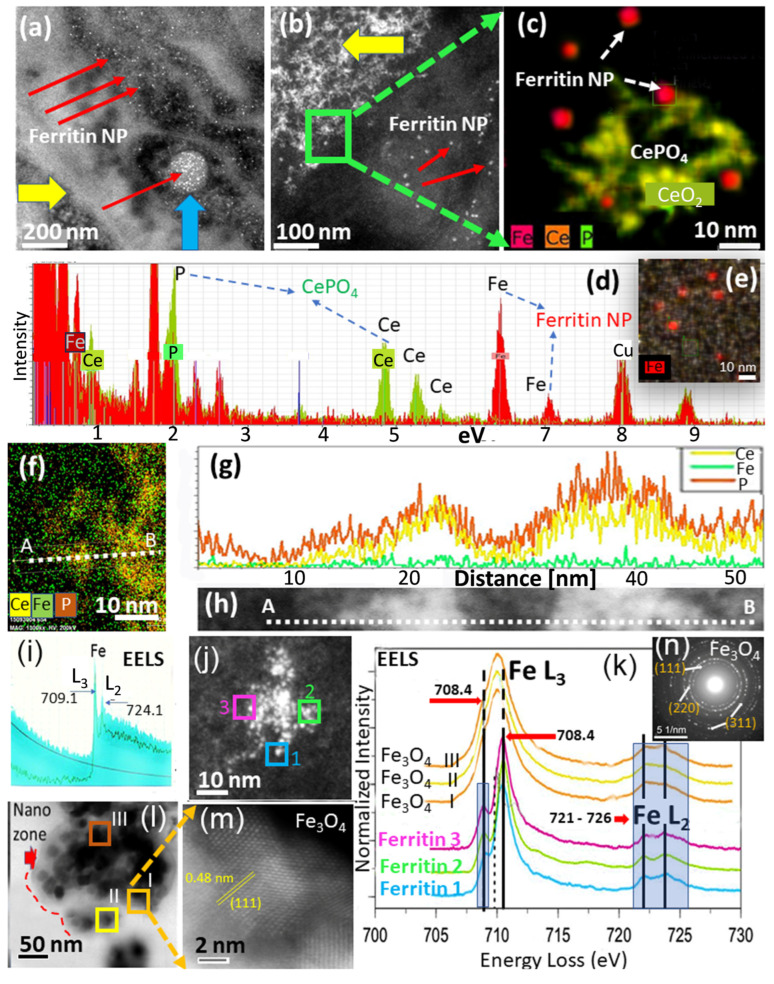
Ferritin and Fe_3_O_4_ particle NPs in human M1-type cells. STEM images of ferritin NPs are shown in (**a**,**b**). Red arrows point to ferritin NPs and yellow arrows to CeO_2_ and CePO_4_ regions. The blue arrow in (**a**) shows a lysosome filled with ferritin NPs. (**c**) is a magnified region from the green square in (**b**), showing ferritin NPs surrounding CePO_4_ nanoneedles that biomineralized from CeO_2_ NPs. (**d**) is an EDS analysis for a ferritin-rich region with an EDS map in (**e**). (**f**–**h**) show EDS map over a CeO_2_/CePO_4_-rich region, with an EDS line scan shown in (**f**) taken across the particles shown in high-res STEM (**h**) and corresponding EDS concentration profile for Ce, P, and Fe shown in (**g**). (**i**) EELS analysis of a ferritin NP with Fe L_3_ and L_2_ edges shown. (**j**) STEM image of a cluster of ferritin NPs with blue, green, and pink squares where the ferritin 1, 2, and 3 EELS analyses shown in (**k**) are taken. (**l**) STEM image of larger Fe_3_O_4_ particle associations within a nanozone region (red dotted line and arrow). The orange square marks the magnified region shown in the STEM image (**m**) with the {111} face of Fe_3_O_4_ marked in yellow. (**l**) orange, yellow, and brown squares from which EELS analyses (I, II, and III) are shown in (**k**). The electron diffraction of the Fe_3_O_4_ particle in (**m**) is shown in (**n**) with additional crystal faces {220} and {311}.

**Table 1 nanomaterials-13-02298-t001:** Nucleus size of M0, M1, and M2 RAW cells after CeO_2_ NP and Ce-ion uptake. Six h results are above the “-” marks; 24 h results are below the “-” marks. Mean sizes (in µm) are in the parentheses below the treatment conditions. NS, *, **, **** indicate nonsignificant and statistically significant differences at *p* < 0.05, 0.01, and 0.0001, respectively, by ANOVA followed by Tukey’s multiple comparisons tests.

6 h	M0 Control (19.44)	M0 CeO_2_ NPs (22.42)	M0 Ce-Ion (19.81)	M1 Control (32.22)	M1 CeO_2_ NPs (30.03)	M1 Ce-Ion (35.71)	M2 Control (18.40)	M2 CeO_2_ NPs (17.58)	M2 Ce-Ion 24.04)
M0 control	-	NS	NS	****	****	****	NS	NS	NS
M0 CeO_2_ NPs	NS	-	NS	****	*	****	NS	NS	NS
M0 Ce-ion	NS	NS	-	****	***	****	NS	NS	NS
M1 control	****	****	****	-	NS	NS	****	****	**
M1 CeO_2_ NPs	NS	NS	NS	****	-	NS	****	****	NS
M1 Ce-ion	****	****	****	NS	****	-	****	****	****
M2 control	NS	NS	NS	****	NS	****	-	NS	NS
M2 CeO_2_ NPs	****	NS	NS	****	****	****	NS	-	*
M2 Ce-ion	*	NS	NS	****	*	****	NS	NS	-
24 h	M0 control (20.88)	M0 CeO_2_ NPs (18.35)	M0 Ce-ion (18.37)	M1 control (28.89)	M1 CeO_2_ NPs (21.61)	M1 Ce-ion (30.19)	M2 control (18.60)	M2 CeO_2_ NPs (15.76)	M2 Ce-ion (17.39)

## Data Availability

Not applicable.

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
