# Peer review of "Macrophage Polarization Status Impacts Nanoceria Cellular Distribution but Not Its Biotransformation or Ferritin Effects"

_nanomaterials, 2023, doi:10.3390/nano13162298_

Round 1

Reviewer 1 Report

Graham et al report that M1 and M2 macrophages internalize nanoceria differently but show identical biotransformation to cerium phosphate nanoneedles and matching co-precipitation of ferritins with nanoceria inside both phenotypes. These findings emphasize the need for evaluating ferritin biomineralization in studies that involve the internalization of nano objects.

The manuscript is of interest for the audience of Nanomaterials.

The reviewer has a few comments :

- what is the hydrodynamic diameter of CeO2 NPs in after the solvothermal synthesis ?

- does this value changes in cell culture medium ?

- is it possible to synthesize larger CeO2 NPs ?

- if yes could it be interesting to test on M1 and M2 macrophages ?

- instead of citrate coating, is it possible to graft other molecules on CeO2 NPs ?

- does the citrate coating impact CeO2 NPs internalization in M1 and M2 macrophages ?

Author Response

Reviewer comments are introduced by a -. Author responses follow.

- what is the hydrodynamic diameter of CeO2 NPs in after the solvothermal synthesis ?

The hydrodynamic diameter of the CeO2 NPs of this study, after dialysis against 0.11 M pH 7.4 citric acid and then DI water, and dispersed in DI water, was 10.8 nm, as we stated in the MS (lines 167 and 214). We added to the revised MS a citation to the reference in which this was reported (Hancock et al, 2021, reference 53 of this report).

- does this value changes in cell culture medium ?

We did not measure the hydrodynamic diameter of the CeO2 NPs of this study in cell culture medium. One would expect there would be interactions between the proteins in the fetal bovine serum of the cell culture medium and the CeO2 NPs, that would change the CeO2 NP’s hydrodynamic diameter. Kumari et al (International Journal of Toxicology 2014;33:86-97) reported their 25 nm CeO2 NPs agglomerated in DMEM cell culture medium to 270 nm.  To measure the hydrodynamic diameter of the CeO2 NPs in the presence of the proteins of the cell culture medium would not be easy, given the proteins would contribute to the dynamic light scattering results and it would be difficult to differentiate the contributions of the CeO2 NPs from the proteins.

  - is it possible to synthesize larger CeO2 NPs ?

Yes. For example, we prepared 5, 15, 30, and 55 nm CeO2 NPs and investigated their biodistribution and biopersistence in rats (Yokel et al, Nanomedicine: Nanotechnology, Biology, and Medicine, 2013;9: 398–407). Kuchibhatla et al (Journal of Materials Research 2019;34:465–473) studied 3 nm CeO2 NPs that they prepared and ~ 10-20 nm particles purchased from Alfa Aesar and ~ 50nm particles purchased from Nanostructured and Amorphous Materials Inc. We could cite many other studies of various sizes of CeO2 NPs. In the present study we investigated ~ 4 nm CeO2 NPs because CeO2 NPs of this size have better anti-oxidant properties than larger CeO2 NPs and are, therefore, more useful in the many pharmaceutical applications that have been demonstrated with CeO2 NPs. ~ 4 nm CeO2 NPs are more likely to be intentionally introduced into mammalian, including human, organisms, than larger CeO2 NPs.    

- if yes could it be interesting to test on M1 and M2 macrophages ?

Yes, the cellular distribution of larger M1 and M2 macrophages could be different than we documented with the ~ 4 nm CeO2 NPs. Of course, that would be another study.

- instead of citrate coating, is it possible to graft other molecules on CeO2 NPs ?

Yes. One common “coating” for nanoparticles is polyethylene glycol (PEG) which mimics cell’s glycocalyx to create “stealth” nanoparticles that are not recognized by macrophages, so circulate longer. Karakoti et al published the synthesis of PEG-coated CeO2 NPs (Journal of the American Chemical Society 2009;131:14144-14145). Heckman et al (ACS Nano 2013; 23:10582-10596) used a citrate/EDTA coating to stabilize their CeO2 NPs. Wu et al (The Journal of Physical Chemistry B, 2002;106: 4569-4577 synthesized cetyltrimethylammonium bromide-coated CeO2 NPs. Dextran-coated CeO2 NPs were prepared by Alpaslan et al (ACS Biomaterials Science and Engineering 2015;1;1096-1103).

- does the citrate coating impact CeO2 NPs internalization in M1 and M2 macrophages ?

We do not know but expect it does. We used citrate-coated CeO2 NPs because citrate stabilizes the CeO2 NPs against agglomeration. The absence of citrate coating would be expected to result in larger (agglomerated) CeO2 NPs, introducing another variable that would have to be studied.

Reviewer 2 Report

The authors conducted important data. This is an interesting paper. The present version of the manuscript is very well developed. The data presented are new and relevant; therefore, the manuscript is suitable for publication in its current form.

Author Response

Reviewer Comments and Suggestions for Authors: The authors conducted important data. This is an interesting paper. The present version of the manuscript is very well developed. The data presented are new and relevant; therefore, the manuscript is suitable for publication in its current form.

Thank you. No responses were required to address these comments.

Reviewer 3 Report

Interaction and biotransformation of intracellular nanoparticles is a complex and critical process. The paper focuses on the important role of macrophages in the clearance of nanoparticles and the effects of nanoparticles on macrophages, including alterations in cellular polarization and immune responses. The mechanism of interaction and biotransformation between CeO2NPs and macrophages (human and murine) was studied by using techniques such as high-resolution electron microscopy. Overall, the material selection is novel, the design is interesting, and the experiments and data analysis seem to have been carried out carefully. However, there are still some problems in this paper, including structural arrangement and manuscript quality, which should be revised.

1. The title is too long and complex to accurately summarize the research content. Especially the logical relationship and specific meaning between "distinct translation" and "the same biotransformation and ferritin effects" are not clear enough, which makes the expression of the entire title not precise and clear enough. It is recommended that the title be rewritten

2. The abstract part seems to lack a clear description of the research purpose, methods, results and conclusions, and the language expression is not concise and clear enough. It is recommended to rewrite it.

3. The connections and logical order between sentences and paragraphs in the introductory paragraph are not clear enough. Suggested reorganization and revision of the introduction to ensure coherence and logical fluency between paragraphs. 

4. The subheadings in the results section do not accurately describe what is involved. It is recommended that the subheadings in the Results section be revisited to ensure that the content of each subsection is accurately summarized.

5. In the "Analytical Electron Microscopy" section, it is mentioned that the main study is "how CeO2 NPs internalized and interacted with differently polarized human macrophages ", but we do not see the effect of CeO2 nanoparticles on macrophages in this section, which should be added.

6. The quality of manuscripts is poor, and there are many problems, such as blurred pictures, chaotic picture typesetting, lack of scale in some pictures, and so on. Please check and correct these problems carefully to ensure the clarity of the image and the cleanliness of the layout.

Author Response

Reviewer comments are followed by the author responses

Interaction and biotransformation of intracellular nanoparticles is a complex and critical process. The paper focuses on the important role of macrophages in the clearance of nanoparticles and the effects of nanoparticles on macrophages, including alterations in cellular polarization and immune responses. The mechanism of interaction and biotransformation between CeO2NPs and macrophages (human and murine) was studied by using techniques such as high-resolution electron microscopy. Overall, the material selection is novel, the design is interesting, and the experiments and data analysis seem to have been carried out carefully. However, there are still some problems in this paper, including structural arrangement and manuscript quality, which should be revised.

  1. The title is too long and complex to accurately summarize the research content. Especially the logical relationship and specific meaning between "distinct translation" and "the same biotransformation and ferritin effects" are not clear enough, which makes the expression of the entire title not precise and clear enough. It is recommended that the title be rewritten.

We revised the title to: Macrophage polarization status impacts nanoceria cellular distribution but not its biotransformation or ferritin effects

  1. The abstract part seems to lack a clear description of the research purpose, methods, results and conclusions, and the language expression is not concise and clear enough. It is recommended to rewrite it.

The Abstract has been rewritten to: The innate immune system is the first line of defense against external threats through the initiation and regulation of inflammation. Macrophage differentiation into functional phenotypes influences the fate of nanomaterials taken up by these immune cells. High-resolution electron microscopy was used to investigate the uptake, distribution, and biotransformation of nanoceria in human and murine M1 and M2 macrophages in unprecedented detail. We found that M1 and M2 macrophages internalize nanoceria differently. M1-type macrophages predominantly sequester nanoceria near the plasma membrane whereas nanoceria is more uniformly distributed throughout M2 macrophage cytoplasm. In contrast, both macrophage phenotypes show identical nanoceria biotransformation to cerium phosphate nanoneedles and simultaneous nanoceria ferritin co-precipitation within the cells. Ferritin biomineralization is a direct response to nanoparticle uptake inside both macrophage phenotypes. We also found the same ferritin biomineralization mechanism occurs after the uptake of Ce-ions into polarized macrophages and into unpolarized human monocytes and murine RAW 264.7 cells. These findings emphasize the need for evaluating ferritin biomineralization in studies that involve the internalization of nano objects, ranging from particles to viruses to biomolecules, to gain greater mechanistic insights into the overall immune responses to nano objects. 

  1. The connections and logical order between sentences and paragraphs in the introductory paragraph are not clear enough. Suggested reorganization and revision of the introduction to ensure coherence and logical fluency between paragraphs.

Changes have been made to the introductory paragraphs to address this point.

  1. The subheadings in the results section do not accurately describe what is involved. It is recommended that the subheadings in the Results section be revisited to ensure that the content of each subsection is accurately summarized.

The content of each section was compared to its subheading, and the subheading revised when appropriate.

  1. In the "Analytical Electron Microscopy" section, it is mentioned that the main study is "how CeO2 NPs internalized and interacted with differently polarized human macrophages ", but we do not see the effect of CeO2 nanoparticles on macrophages in this section, which should be added.

The section “Analytical Electron Microscopy” presents and discusses results of CeO2 NPs internalization, and effects of macrophages on the internalized NPs. The CeO2 nanoparticle effects on the macrophages is presented and discussed in the section “Ferritin Biomineralization”. To address this reviewer’s comment the first sentence of the “Analytical Electron Microscopy” section has been revised to “How CeO2 NPs internalized into differently polarized human macrophages is detailed in Figure 2.”

  1. The quality of manuscripts is poor, and there are many problems, such as blurred pictures, chaotic picture typesetting, lack of scale in some pictures, and so on. Please check and correct these problems carefully to ensure the clarity of the image and the cleanliness of the layout.

Nanomaterials’ Instructions for Authors request that figures be included in the manuscript, which we did. To provide better quality figures we are submitting the figures as separate high-res (300 dpi) files (in addition to their inclusion in manuscript). The legends for all figures have been edited. Please use the high-res images in publication of both the article and the Supporting Information.

Round 2

Reviewer 3 Report

Accepted